# The Effectiveness of Pelvic Floor Muscle Training in Men after Radical Prostatectomy Measured with the Insert Test

**DOI:** 10.3390/ijerph19052890

**Published:** 2022-03-02

**Authors:** Dorota Szczygielska, Andrzej Knapik, Teresa Pop, Jerzy Rottermund, Edward Saulicz

**Affiliations:** 1College of Medical Sciences, Institute of Health Sciences, University of Rzeszow, ul. Warzywna 1a, 35-310 Rzeszów, Poland; tpop@ur.edu.pl; 2Department of Adapted Physical Activity and Sport, School of Health Sciences in Katowice, Medical University of Silesia in Katowice, ul. Medyków 12, 40-751 Katowice, Poland; aknapik@sum.edu.pl; 3Health and Social Work, St. Elizabeth University, Namestie 1, maja 1, 811 02 Bratislava, Slovakia; jerzy_rottermund@op.pl; 4Department of Kinesiotherapy and Special Methods in Physiotherapy, The Jerzy Kukuczka Academy of Physical Education, ul. Mikołowska 72a, 40-065 Katowice, Poland; e.saulicz@awf.katowice.pl

**Keywords:** urinary incontinence, radical prostatectomy, pelvic floor muscle training, 1 h pad weighing test

## Abstract

A commonly used physiotherapeutic method for the treatment of urinary incontinence (UI) after radical prostatectomy (RP) is pelvic floor muscle training (PFMT). The aim of this study was to evaluate the effectiveness of PFMT by enhanced biofeedback using the 1h pad-weighing test. The following factors were taken into consideration in the analysis of PFMT effectiveness: the relevance of the patients’ age, time from RP, BMI, mental health, functional state, and depression. A total of 60 post-RP patients who underwent 10-week PFMT were studied. They were divided into groups: A (*n* = 20) and B (*n* = 20) (random division, time from RP: 2–6 weeks) and group C (time from RP > 6 weeks). Group B had enhanced training using EMG biofeedback. UI improved in all groups: A, *p* = 0.0000; B, *p* = 0.0000; and C, *p* = 0.0001. After the completion of PFMT, complete control over miction was achieved by 60% of the patients in group A, 85% in group B, and 45% in group C. There was no correlation between the results of PFMT efficacy and patients’ age, BMI, time from RP, mental health, functional state, and depression. PFMT is highly effective in UI treatment. The enhancement of PFMT by EMG biofeedback seems to increase the effectiveness of the therapy.

## 1. Introduction

Prostate cancer (PCa) is the second most common cancer type in men after lung cancer. It ranks fifth globally among the causes of death. GLOBOCAN estimates from 2020 speak of over 1,400,000 new cases globally [1]. PCa is the fifth leading cause of death, posing a global public health challenge [2]. Radical prostatectomy (RP) is a recognized and effective method of treatment for PCa [3,4]. It is especially effective at an early stage of clinical advancement. Research shows that the ten-year survival rate for patients after RP is 92% [5].

RP involves the removal of the entire prostate between the urethra and the bladder, and resection of the seminal vesicles with an appropriate amount of surrounding tissue in order to obtain negative surgical margins [6]. Often, during this operation, pelvic lymph nodes are also removed on both sides. Of course, nerve-sparing surgery should be performed in all patients where possible [7]. For this purpose, various surgical techniques are used, and their consequences are constantly being researched [6,8]. The main goal of RP surgery is to cure neoplastic disease. However, side effects such as urinary incontinence (UI), and erectile dysfunction (impotence) may significantly reduce the quality of life [9,10].

UI is a significant health problem that has physical, social, and economic ramifications for patients as well as for the wider community [11]. It limits daily activity due to urine leakage and negatively affects sexual and interpersonal relations, social interactions, and mental well-being [12]. Identified risk factors for UI after RP are age, bilateral neurovascular bundle resection, and anastomotic stricture [13]. Despite the improvement of surgical methods and the use of modern techniques, as well as drug treatment, the problem of UI in people after RP procedures has not been eliminated thus far [14,15,16]. The data on UI indicators are very divergent. This is largely due to differences in the definitions and methodologies used to measure urinary incontinence [17]. There is still no definition of standards for the collection of UI data [18]. Some researchers believe that the criterion for full continence control is the lack of pads for 24 h in the last week. Other researchers believe that ≤1 pad per day means “social continence” [19,20,21]. Regardless of the adopted criteria, researchers report that UI may affect up to 80% of patients after RP [22].

Parallel to the search for more modern RP techniques, effective in the treatment of PCa and at the same time minimizing the risk of UI, other methods used to help patients in need of these are being sought. These include the methods of physiotherapy and behavioral therapy [23,24,25]. The assessment of the effectiveness of these methods has thus far varied. There are critical opinions [26], and there are also research results indicating the effectiveness of these interventions [27,28]. One commonly used method is pelvic floor muscle training (PFMT). According to studies, enhancing PFMT may increase its effectiveness [29]. It was assumed that pad-weighing tests (PAT) [30] would be a good way to assess the effectiveness of PFMT. We decided to investigate the effect of age, time from RP, and PFMT enhancement on the intensity of UI symptoms.

## 2. Materials and Methods

### 2.1. Participants

The study included 60 post-RP men aged 51–75 years old (mean: 63.60; SD: 6.21). They were residents of Rzeszów and the surrounding area (south-eastern Poland). The inclusion criteria were: RP surgery performed with the radical retropubic prostatectomy method; time from RP: minimum 2 weeks (removal of the catheter and urinary continence control); diagnosis of UI by a urologist specialist; patient consent to participate in the PFMT program; and positive results on the following scales: The Abbreviated Mental Test Score (AMTS) [31], Index of Activities for Daily Living (IADL) [32], and Geriatric Depression Scale (GDS) [33]. AMTS is a short 10-point test used to assess the possibility of developing dementia. The purpose of the IADL application is to define the life independence of the examined people. The GDS, on the other hand, is a screening scale used to identify symptoms of depression. The use of these scales was to ensure the highest possible effectiveness of PFMT.

All subjects were divided into three groups. There were a total of 40 patients in groups A and B, whose time since RP was less than 6 weeks (mean: 27 days). The division into these groups was random. Group C consisted of 20 patients, whose period from RP was greater than 6 weeks (Table 1).

### 2.2. Program of Pelvic Floor Muscle Training

The experiment program for all groups lasted 10 weeks. All the subjects exercised once a week in a physiotherapy office under the supervision of a physiotherapist specializing in urological physiotherapy. Each of the patients participating in the experiment was also recommended to repeat 10–15 min of home practice sessions three times a day (in the morning, at noon, and in the evening). The duration of the therapeutic sessions in each group is presented in Table 2.

In all the groups, the exercises consisted of 10 short 1 s tensions of the pelvic floor muscles and 10 long tensions, lasting 10 s. This sequence of exercises was performed in 3 body positions: in the supine position, in a sitting position on a stool, and in a standing position. In order to control the muscle tone of the pelvic floor, the tension of the muscles acting synergistically with the muscles of the pelvic floor, i.e., the gluteal muscles, thigh adductors, and abdominal muscles, was gradually deactivated in each position. Each patients, apart from repeating the program of learned exercises three times a day, was supposed to interrupt the flow of urination during voiding once a day and continue voiding again. The aim of this action was to control the ability to tone the pelvic floor muscles, especially the outer part of the urethral sphincter.

In group B, the exercises based on EMG biofeedback were used in addition to the method of re-education of the pelvic floor muscles, which was analogous to the other groups. For this purpose, a four-channel NORAXON camera was used. The aim was to learn controlled tension in the so-called local stabilizers of the lumbar spine: the multisection muscles and the transverse abdominal muscle. This is because the tension of these muscles is accompanied by co-contraction of the pelvic floor muscles [34]. Superficial electrodes were placed paraspinally on both sides of the body, one pair at the height of the posterior superior iliac spines and the other pair at the level of the iliac plate. This location of the electrodes allowed for the collection of biopotentials from the multi-divided muscle. Similar exercises to those in the pelvic floor muscle training were performed in the case of activation of the so-called local stabilizers with 10 short voltages lasting 1 s and 10 long voltages each lasting 10 s.

### 2.3. One-Hour Pad-Weighing Test

Before starting the exercise program and immediately after its completion, a one-hour pad-weighing test (PAT) was performed in all the subjects (PAT1 and PAT2) [35]. The procedure for conducting the PAT was as follows:Patient urinates prior to test.Measurement of the weight of the insole (measurement accuracy: ±1 g).Insertion of the insole—starting the test.Drinking 500 mL of sodium-free liquids by the examined person within a short time (up to 15 min).Walking by the subject for 30 min; while walking, the subject must cover one floor upstairs and downstairs.Performing the activities: 10 times rising from a sitting position, 10 times coughing, 5 times lifting a small object from the ground, 1 min jogging or walking in a place, and washing hands for 1 min under running cold water.Removal of the insert and measurement of its weight.

The insert test allows one to evaluate the volume of uncontrolled leakage of urine over a period of time. A weight of 1 g was assumed to be equivalent to 1 mL of urine. Measurements were made using an electronic balance. Accuracy of the measurement: 0.01 g.

The experiment design was approved by the Bioethics Committee of the University of Rzeszów no. 2/11/2009.

### 2.4. Statistical Analysis

Non-parametric statistics were used. The Wilcoxon pairwise test was used to evaluate the effectiveness of the experiment and the comparison of PAT 1 and PAT2. Comparisons between groups were made using the Kruskal–Wallis ANOVA test. The relationships between the variables were calculated using the Spearman correlation. The adopted level of statistical significance was *p* < 0.05.

## 3. Results

The comparison of PAT 1 (before PFMT) and PAT 2 (after PFMT) shows a statistically significant decrease in all three groups, indicating the effectiveness of PFMT. The differences were as follows: group 1: *p* = 0.0000, group 2: *p* = 0.0000, and group 3: *p* = 0.0001. Significance in group B seems to be more pronounced than in groups A and C (Figure 1). However, the intergroup comparisons showed no differences in PAT 1 and PAT 2, but there were differences in PAT 2–PAT 1. This allowed us to treat all subjects as a homogeneous group in terms of measurement both before and after the experiment. However, the size of the standard deviations and confidence intervals indicate a large individual variability among the respondents (Table 3).

The detailed analyses showed that in only two subjects (3%) was there no difference in PAT2–PAT 1, and in the other two, the difference was less than 1 mL. On the other hand, assuming the criterion of total voiding control in PAT 2 (urine loss during the test of less than or equal to 1 g [36]), this was found in group A in 12 patients (60%) with an average decrease of 2 mL, and in group B in 17 patients (85%) with an average decrease of 1.73 mL, while it was found in group C in 9 subjects (45%) with an average decrease of 2.02 mL.

The correlation analysis performed did not show any correlation between the age of the respondents, BMI, time since RP, AMTS, IADL, and GDS, nor any differences between PAT 2 and PAT 1.

## 4. Discussion

The latest epidemiological data on PCa in the European Union (EU) indicate 10 cases per 100 thousand. This is a decline in the incidence of over 7% since 2015, which is a positive trend. Unfortunately, the exception to this trend, favorable from the point of view of public health, is Poland, where an upward trend was recorded, with an increase of 18%. The statistics show that mortality from PCa is decreasing. This trend is expected to continue [37]. Actions to combat PCa go in four directions: further identification of risk factors [38], related prevention [39,40], effective early methods of diagnosis [41], and possible effective forms of treatment [42]. At the same time, methods for the treatment of the main complications after RP, namely, erectile dysfunction and UI, are also being sought [43]. Various methods are used to help patients, mainly consisting of the use of PFMT and enhancing its effectiveness through the use of electrostimulation [25,44]. This is very important for the quality of life of patients as well as their families [45,46].

The results of the presented studies show the effectiveness of PFMT in patients after RP. There were statistically significant differences between PAT 1 and PAT 2 in all three groups. This seems to emphasize the importance of biofeedback in this therapy, which is consistent with previous observations [27,46,47].

Researchers’ views on the effectiveness of PFMT differ. Some researchers emphasize the benefits of such training and the importance of training before surgery [48,49]. They believe that such training can effectively affect muscle control after RP. On the other hand, others question the validity of PFMT as first-line rehabilitation, claiming that UI symptoms disappear with time, regardless of the procedure [50]. Conclusions from Hall’s review of 108 studies regarding this problem state that the causes of these discrepancies are the varied content of PFMT programs, the quality of reporting, the descriptions of exercise positions, and disproportion in the muscles subject to interventions [51]. In the results presented here, the authors tried to precisely describe the applied PFMT program. On the other hand, the low percentage of respondents with full voiding control in group C contradicts the views on the spontaneous resolution of UI symptoms and suggests starting PFMT as early as possible after RP.

Too weak a synergy between PFM and other muscles is believed to be one of the reasons for the non-optimal effectiveness of PFMT [52]. This mainly applies to the musculus transversus abdominis, as well as the musculus gluteus major and the musculus adductor femoris. The reasons can be complex. According to Sapsford, one can make an analogy between men and postpartum women. As a result of the local trauma related to RP, there may be deficiencies in muscle recruitment despite pain relief, and the neurological deficit in co-contraction results in loss of control over urinary continence. This control depends on both the tonic and phasic effects of PFM: bladder stability, higher resting urethral pressure than bladder pressure, and adequate regulation of urethral pressure during strong stimuli (coughing and sneezing) and exertion. Stress urinary incontinence (SUI) is the most common presentation following RP [53]. Factors disturbing this mechanism should also be taken into account, such as acute low back pain, fascia laxity, or smooth muscle dysfunctions [34]. Therefore, “To improve a specific performance by strengthened musculature, the muscles must be trained with movements as close as possible to the desired movement or actual skill” [54]. This formed the basis for the preparation and execution of the experiment presented here. This was also a justification for the application of the verification of its effects: a one-hour PAT. The assessment of the effectiveness of PFMT with the use of a one-hour PAT, classified as a short-term test, may be questionable. Some researchers believe that long-term tests, lasting from 12 to 48 h, have better sensitivity and acceptable repeatability [55]. However, from a practical point of view, one-hour or two-hour tests have some advantages: above all, apart from being non-invasive, PAT have a short test execution time, allowing one to control their performance to some extent, and there is no need to visit the office again after their execution, which significantly reduces disruption for patients (costs and time spent traveling to the examination) [30]. The confirmation of this thesis is their use in various studies related to the UI problem [56,57].

## 5. Conclusions

The scale of the UI problem and the above-mentioned consequences seem to justify the implementation of PFMT physiotherapeutic procedures as soon as possible after RP. Their use, as judged by the one-hour PAT, is effective. PFMT is probably more effective after biofeedback. The one-hour PAT is a practical tool to be used in the offices of physiotherapists dealing with the UI problem, but the repeatability of this test should be confirmed in further research.

## Figures and Tables

**Figure 1 ijerph-19-02890-f001:**
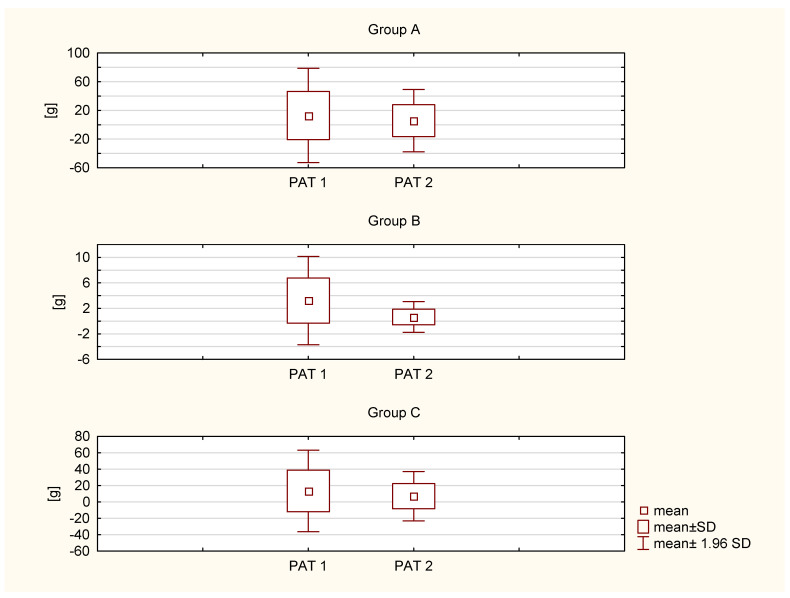
Comparison of PAT 1 and PAT 2 in individual groups.

**Table 1 ijerph-19-02890-t001:** Characteristics of the groups.

Variable	Group						*p*
	A		B		C		
	Mean (SD)	±95%CI	Mean (SD)	±95%CI	Mean (SD)	±95%CI	
Age	64.15 (5.80)	61.44–66.86	63.15 (6.43)	60.14–66.16	63.50 (6.65)	60.39–66.61	0.8360
Time from RP	27.55 (7.13)	24.21–30.89	27.10 (9.50)	22.65–31.55	160.05 (131.38)	98.56–221.54	0.0381 ^1^
BMI	28.68 (3.39)	27.10–30.27	26.47 (2.20)	25.44–27.50	27.20 (2.39)	26.09–28.32	0.0000 ^2^
AMTS	9.50 (0.51)	9.26–9.74	9.50 (0.69)	9.18–9.82	9.60 (0.60)	9.32–9.88	0.1546
IADL	5.65 (0.49)	5.42–5.88	6.00		5.90 (0.31)	5.76–6.04	0.0050 ^3^
GDS	6.25 (2.05)	5.29–7.21	4.95 (1.82)	4.10–5.80	5.75 (1.29)	5.15–6.35	0.7469

^1^ A-C: *p* = 0.0000, B-C: *p* = 0.0000; ^2^ A-B: *p* = 0.0124; ^3^ A-B: *p* = 0.0222.

**Table 2 ijerph-19-02890-t002:** Exercise time in individual groups and the use of biofeedback.

Group	Exercise Time in a Physiotherapist’s Office	Biofeedback	Recommendations for Daily Home Exercises
A	15–20 min	no	3 × 10^−15^ min
B	20–30 min	yes	
C	15–20 min	no	

**Table 3 ijerph-19-02890-t003:** Descriptive statistics of PAT 2 and PAT 1 and the PAT difference, divided into groups.

Group	PAT 2			PAT 1			PAT2–PAT 1		
	Mean (SD)	±95%CI	*p*	Mean (SD)	±95%CI	*p*	Mean (SD)	±95%CI	*p*
A	5.75 (22.21)	−4.64–16.14	0.1218	12.95 (33.53)	−2.74–28.64	0.1893	7.20 (13.30)	0.97–13.42	0.2231
B	0.65 (1.23)	0.08–1.22		3.22 (3.54)	1.56–4.88		2.52 (2.85)	1.19–3.85	
C	7.01 (15.35)	−0.18–14.19		13.44 (25.38)	1.56–25.32		6.44 (13.44)	0.15–12.72	

## Data Availability

Not applicable.

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
