# Peer review of "The Effectiveness of Pelvic Floor Muscle Training in Men after Radical Prostatectomy Measured with the Insert Test"

_ijerph, 2022, doi:10.3390/ijerph19052890_

Round 1

Reviewer 1 Report

This study is to assess the effectiveness of PFMT by biofeedback using the one hour pad weighting test. 

I have some questions about this study. 

  1. Whether or not nerve sparing in surgery is very important in urinary incontinence.  It is needed to be add this context in the manuscript.
  2.  It would be good to add whether or not an anti-incontinentic drug such as anti cholinergics.
  3. why do authors choose inclusion criteria "time from RP minimum 2 weeks"? Incontinence after RP was improved spontaneously after 1-2 months. 
  4. It is need to explain the details of AMTS, IADL, and GDS.
  5. Discussion is relatively short. It is good to be add more references for this manuscript.  

Author Response

Prosze zobaczyć w załączniku.

Reviewer 2 Report

This paper investigates the efficacy of the physiotherapy, specifically, Pelvic Floor Muscle Training and its extended training to alleviate the urinary incontinence after the radical prostatectomy. The findings favour the said therapy, but there are a few concerns that come out regarding the statistical analysis.

p.4, Sec 2.4: is it a standard ANOVA test which was used or a non-parametric one-way ANOVA test (Kruskal-Wallis test)? The sample sizes in this study are relatively small, and hence less likely to be normally distributed. It is not clear from the paper whether the data were checked to be normally distributed or not. This section needs to be expanded to include more details and if needed analyses have to be rerun as well with the appropriate test.

Sec. Results: it does not come clear in the paper what the differences are between PAT1 and PAT2? What does PAT2-PAT1 mean? I assume 1 and 2 refer to the number of hours the test was conducted? It's not clear to me anyway. No description of the Y-axis in Fig.1. I would also suggest for the non-significant p-values to be disclosed for reproducibility and better understanding of the results. Again this section needs more work.

It also looks like the therapy might be less effective when undertaken 6 weeks or longer after the RP, only 45% achieved a full control in this group compared to 60% in the less than 6 weeks group, but not much discussion about it in the paper.

Small corrections:

P1,l.31: maybe "spike" or "peak" instead of "speak"?
P2,l.49: maybe "inconsistent" or "varied" instead of "divergent", just a suggestion
P3,l.116: IUD? What does this abbreviation stand for?
P4,l.133: maybe "Our analyses were conducted using ANOVA test", this sentence needs rephrasing. 
l.166: IU or UI?
l.171: remove ".in group B."

Reviewer 3 Report

The authors describes that pelvic floor muscle training (PFMT) is found to be effective in urinary incontinence management especially after radical prostatectomy. The overall work was demonstrated properly in the manuscript and the topic is scholarly adequate. However, the reviewer would like to discuss the followings with the authors.

  1. There are many review papers regarding PFMT assessment in UI treatment. The introduction needs to better address the authors' rationale why their work is discerning from others.  
    1. "Pelvic floor muscle training to improve urinary incontinence after radical prostatectomy: a systematic review of effectiveness" (https://www.ncbi.nlm.nih.gov/books/NBK73486/)
    2. "Reconsideration of pelvic floor muscle training to prevent and treat incontinence after radical prostatectomy" (https://pubmed.ncbi.nlm.nih.gov/31882228/)
  2. Graphical illustrations may be helpful in describing both pelvic floor muscle training (PFMT) and Pad-weighing tests.
  3. In addition, the authors are encouraged to better illustrate what physical activities of PFMT increase its efficacy in UI treatment. 
  4. Optionally, the authors may want to discuss the continuous management of UI symptoms by incorporating a newly emerging concept of "precision health". As in its core, Smart Toilet (e.g., https://pubmed.ncbi.nlm.nih.gov/32251391/) would provide an ideal solution to monitor UI symptoms over a certain period of time. 
  5. (A) IRB approval number(s) should be included in the Institutional Review Board Statement section.  

Round 2

Reviewer 1 Report

Thank your for your responses.

I suggest that this study is ready to be published in this journal. 

Reviewer 2 Report

A few minor corrections:

1) It would be more sound to use the scientific notation for the p-values, the degree is important.

2) l.139. Wilcoxon test?

3) l.141. Maybe better to add "test" after ANOVA?

3) Fig.1 needs to be improved visually. Groups at the top of the figure are A, 2 and 3? Should not they be A, B and C? They are also not consistently aligned, same with the X-axis labels, they do not look appropriately aligned.

Reviewer 3 Report

It seems that the authors wish to reflect most of the reviewer's concerns in a future work. Considering the scope of the journal - environmental and public health, the authors need to better illustrate their methodologies for general audience of the journal. The authors are encouraged at least to find proper references to what the reviewer suggested and to cite them in the manuscript. For example, graphical illustrations for PFMT can be found in any online sources and the authors may want to include this.  
